

# High-dimensional and permutation invariant anomaly detection

**Vinicius Mikuni**[1][⋆] **and Benjamin Nachman**[2,3][†]

**1** National Energy Research Scientific Computing Center,
Berkeley Lab, Berkeley, CA 94720, USA
**2** Physics Division, Lawrence Berkeley National Laboratory, Berkeley, CA 94720, USA
**3** Berkeley Institute for Data Science, University of California, Berkeley, CA 94720, USA

⋆ vmikuni@lbl.gov , † bpnachman@lbl.gov

## Abstract

Methods for anomaly detection of new physics processes are often limited to low-dimensional spaces due to the difficulty of learning high-dimensional probability densities. Particularly at the constituent level, incorporating desirable properties such as permutation invariance and variable-length inputs becomes difficult within popular density estimation methods. In this work, we introduce a permutation-invariant density estimator for particle physics data based on diffusion models, specifically designed to handle variable-length inputs. We demonstrate the efficacy of our methodology by utilizing the learned density as a permutation-invariant anomaly detection score, effectively identifying jets with low likelihood under the background-only hypothesis. To validate our density estimation method, we investigate the ratio of learned densities and compare to those obtained by a supervised classification algorithm.



# 1 Introduction

Anomaly detection (AD) has emerged as a complementary strategy to classical model-dependent searches for new particles at the Large Hadron Collider and elsewhere. These tools are motivated by the current lack of excesses and the vast parameter space of possibilities [1, 2]. Machine learning (ML) techniques are addressing these motivations and also allowing for complex particle physics data to be probed holistically in their natural high dimensionality [3].

Nearly all searches for new particles begin by positing a particular signal model, simulating the signal and relevant Standard Model (SM) backgrounds, and then training (with or without ML) a classifier to distinguish the signal and background simulations. Machine learning–based AD tries to assume as little as possible about the signal while also maintaining the ability to estimate the SM background. Two main classes of ML approaches are unsupervised and weakly/semi-supervised. Unsupervised methods use 'no' information about the signal in training while weakly/semi-supervised methods use limited or noisy labels. The 'no' is in quotes because there is often implicit signal information used through event and feature selection.

At their core, unsupervised methods select events that are rare, while weakly/semi supervised methods focus on events that have a high likelihood ratio with respect to some reference(s). The first ML-based AD proposals in high energy physics explored both weakly/semi-supervised classifiers [4–6] as well as unsupervised learning via a type of ML tool called an autoencoder [7–9]. Since that time, there have been many proposals in the literature (see e.g. Ref. [10]), community challenges comparing a large number of approaches [11, 12], and first physics results using a variety of methods [13–18]. Even though a number of weakly supervised methods have statistical guarantees of optimality that unsupervised method lack [19,20], there has been significant interest in unsupervised AD because of its flexibility.

The flexibility of unsupervised learning leads to a number of challenges. There is no unique way to estimate the probability density of a given dataset, with some methods offering only an implicit approximation through proxy quantities like the reconstruction fidelity of compression algorithms. The probability density itself is not invariant under coordinate transformations, so the selected rare events will depend on the feature selection [21]. Even though particle physics data are often described by high- (and variable-)dimensional, permutation-invariant sets ('point clouds'), there has not yet been a proposal to use explicit density estimation techniques for AD that account for all of these properties. Implicit density estimation has been studied with a variety of high-dimensional, but mostly fixed-length representations, such as (variational) autoencoders and related approaches [8,9,22–28]. Similarly, generative models are capable to implicitly learn the the density through data generation. In this case, high quality samples can be created but the exact value of the density is not explicitly known.Since our validation protocol requires access to the density, we focus only on explicit methods. So far, the only[1] high-dimensional explicit density estimators in particle physics [31–36] have been based on normalizing flows [37, 38]. These works process fixed-length and ordered inputs, but recent work has shown with higher-level observables how to accommodate variable-length and permutation invariance with normalizing flows [39].

However, variable-length is not a natural property for normalizing flows which are built on bijective maps from the data space to a fixed-length latent space. In contrast, a newer class of methods called score-matching or diffusion models do not have this restriction. These techniques estimate the gradient of the density instead of the density itself, and therefore have fewer restrictions than normalizing flows. Diffusion models have been shown to accurately

---

[1]Except for Ref. [29,30], which discretize the phase space and turn the problem into a multi-class classification task.

model both high- [40] and/or variable- [41–44] dimensional feature spaces. Despite these early successes, such models have not been used yet for explicit density estimation in particle physics.

We propose to use point cloud diffusion models combined with explicit density estimation for AD. Our approach is based on Ref. [42], and inherits the ability to process variable-length and permutation-invariant sets. From the learned score function, we estimate the data density and provide results for two different diffusion models; one trained with standard score-matching objective and one trained using maximum likelihood estimation. Since the true density is not known, we quantify the performance of the density estimation with likelihood ratios. Finally, we demonstrate the performance of the density as an anomaly score for top quark jets as well as jets produced from dark showers in a hidden valley model. Other tasks that require access to the data density could also benefit from our method.

This paper is organized as follows. Section 2 introduces the methodology of maximum likelihood-based diffusion modeling for permutation-invariant density estimation. The datasets used for our numerical examples are presented in Sec. 3 and the results themselves appear in Sec. 4. The paper ends with conclusions and outlook in Sec. 5.

## 2 Score matching and maximum likelihood training of diffusion models

Score-based generative models are a class of generative algorithms that aim to generate data by learning the score function, or gradients of the logarithm of the probability density of the data. The training strategy presented in Ref. [45] introduces the idea of denoising score-matching, where data can be perturbed by a smearing function and matching the score of the smeared data is equivalent to matching the score of the smearing function Ref. [46]. Given some high-dimensional distribution $\mathbf{x} \in \mathbb{R}^D$, the score function we want to approximate, $\nabla_{\mathbf{x}} \log p_{\text{data}}$, with $\mathbf{x} \sim p_{\text{data}}$, is obtained by minimizing the following quantity

$$\frac{1}{2}\mathbb{E}_t\mathbb{E}_{p_t(\mathbf{x})}\left[\lambda(t)\left\|\mathbf{s}_\theta(\mathbf{x_t}, t) - \nabla_{\mathbf{x_t}} \log p_t(\mathbf{x_t}|x_0)\right\|_2^2\right].\tag{1}$$

The goal of a neural network $\mathbf{s}_\theta(\mathbf{x_t}, t)$ with trainable parameters $\theta$ and evaluated with data $\mathbf{x}_t$ that have been perturbed at time $t$ is to give a time-dependent approximation of the score function. The time dependence of the score function is introduced to address the different levels of perturbation used in each time step. At times near 0, at the beginning of the diffusion process ($\mathbf{x}(0) := \mathbf{x}_0 := \mathbf{x}$), the smearing applied to data is small, gradually increasing as time increases and ensures that at longer time scales the distribution is completely overridden by noise. Similarly, the positive weighing function $\lambda(t)$ can be chosen independently and determines the relative importance of the score-matching loss at different time scales.

The score function of the perturbed data is calculated by using a Gaussian perturbation kernel $p_\sigma(\tilde{x}|x) := \mathcal{N}(\mathbf{x}, \sigma^2)$ and $p_\sigma(\tilde{\mathbf{x}}) := \int p_{\text{data}}(\mathbf{x})p_\sigma(\tilde{\mathbf{x}}|\mathbf{x})d\mathbf{x}$, simplifying the last term of Eq. 1

$$\nabla_{\tilde{\mathbf{x}}} \log p_\sigma(\tilde{\mathbf{x}}|\mathbf{x}) = \frac{\mathbf{x} - \tilde{\mathbf{x}}}{\sigma^2} \sim \frac{\mathcal{N}(0, 1)}{\sigma}.\tag{2}$$

The learned approximation to the score function can then be used to recover the data probability density by solving the following equation:

$$\log p_0(\mathbf{x}_0) = \log p_T(\mathbf{x}_T) + \int_0^T \nabla \cdot \tilde{\mathbf{f}}_\theta(\mathbf{x}_t, t)dt,\tag{3}$$

with

$$\tilde{\mathbf{f}}_\theta(\mathbf{x}_t, t) = \left[ f(t)\mathbf{x}_t - \frac{1}{2} g(t)^2 s_\theta(\mathbf{x}_t, t) \right]. \tag{4}$$

The drift ($f$) and diffusion ($g$) coefficients are associated with the parameters of the Gaussian perturbation kernel. In our studies, we use the VPSDE [47] framework with velocity parameterization as used in [42]. In this parameterization, the score function of the perturbed data reads:

$$s_\theta(\mathbf{x}_t, t) = \mathbf{x}_t - \frac{\alpha_t}{\sigma_t} \mathbf{v}_\theta(\mathbf{x}_t, t), \tag{5}$$

where the outputs of the network prediction, $\mathbf{v}_\theta(\mathbf{x}_t, t)$, are combined with the perturbed data, $\mathbf{x}_t$, and the mean and standard deviation of the induced perturbation kernel $\mathcal{N}(\mathbf{x}(0)\alpha, \sigma^2)$. A cosine schedule is used with $\alpha_t = \cos(0.5\pi t)$ and $\sigma_t = \sin(0.5\pi t)$. The resulting drift and diffusion coefficients are also identified based on the perturbation parameter as

$$\begin{aligned} f(\mathbf{x}, t) &= \frac{d \log \alpha_t}{dt} \mathbf{x}_t, \\ g^2(t) &= \frac{d\sigma_t^2}{dt} - 2\frac{d \log \alpha_t}{dt} \sigma_t^2. \end{aligned} \tag{6}$$

While the estimation of the data probability density is independent from the choice of the weighing function $\lambda(t)$ described in Eq. 1, different choices can enforce different properties to the learned score function. For example, the velocity parameterization in Eq. 5 implicitly sets $\lambda(t) = \sigma(t)^2$, which avoids the last ratio in Eq. 2 that diverges as $\sigma(t) \to 0$ at times near 0. On the other hand, Ref. [48] shows that choosing $\lambda(t) = g(t)^2$ turns the training objective in Eq. 1 into an upper bound to the negative log-likelihood of the data, effectively allowing the maximum likelihood training of diffusion models and possibly leading to more precise estimates of the data probability density. The negative aspect of this choice is that the lack of the multiplicative $\sigma^2$ term can lead to unstable training. This issue can be mitigated by using an importance sampling scheme that reduces the variance of the loss function. During the training of the likelihood weighted objective we implement the same importance sampling scheme based on the log-SNR implementation defined in [49] where the time parameter is sampled uniformly in $-\log(\alpha^2/\sigma^2)$ while in the standard implementation the time component itself is sampled from an uniform distribution.

## 3 Top quark tagging dataset and semi-visible jets

The top quark tagging dataset is the widely-used community standard benchmark from Ref. [50,51]. Events are simulated with Pythia 8 [52,53] and Delphes [54,55] (ATLAS card). The background consists of dijets produced via Quantum Chromodynamics (QCD) and the signal is top quark pair production with all-hadronic decays. The default energy flow algorithm in Delphes is used to create jet consituents, which are clustered using the anti-$k_T$ algorithm with $R = 0.8$. [56–58]. All jets in the range $550 \text{ GeV} < p_T < 650 \text{ GeV}$ and $|\eta| < 2$ are saved for processing. Each jet is represented with up to 100 constituents (zero-padded if fewer; truncated if more).

In practice, supervised learning should be used to look for top quark jets.[2] To illustrate the anomaly detection abilities of our approach, we also simulate jets produced from a dark shower within a hidden valley model [59–62]. Our dark showers are motivated by[3] Ref. [63],

---

[2]Top quark jet modeling has known inaccuracies, so there still may be utility in training directly with (unlabeled) data, but since it is possible to isolate relatively pure samples of top quark jets in data, this is far from 'anomaly detection'.

[3]In contrast to Ref. [63], our mesons have much higher masses, which makes the substructure more non-trivial.

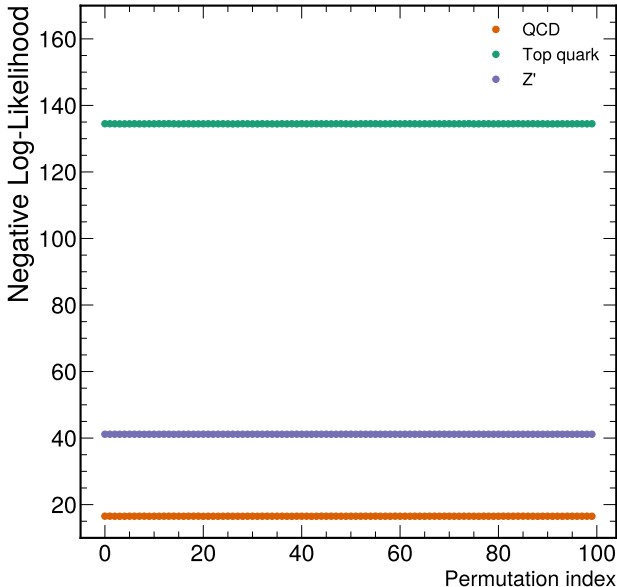

Figure 1: Estimated negative log-likelihood in the model trained exclusively on QCD jets, evaluated on a single jet under multiple permutations of the input particles.

and consist of a $Z'$ with a mass of 1.4 TeV that decays to two dark fermions charged under a strongly coupled U(1)'. These fermions have a mass of 75 GeV and hadronize into dark pion and $\rho$ mesons, each of which can decay back to the Standard Model. The meson masses are 150 GeV, resulting in two-prong jet substructure. The anomaly detection performance evaluated in the dataset similar to Ref. [63] is reported in App. B.

## 4 Results

The network implementation and training scheme used to train the diffusion model are the same ones introduced in Ref. [42], based on the DEEPSETS [64] architecture with Transformer layers [65]. This model is trained to learn the score function of the jet constituents in $(\Delta\eta, \Delta\phi, \log(1 - p_{Trel}))$ coordinates, with the relative particle coordinates $\Delta\eta = \eta_{part} - \eta_{jet}$, $\Delta\phi = \phi_{part} - \phi_{jet}$, and $p_{Trel} = p_{Tpart}/p_{Tjet}$ calculated based on the jet kinematic information. The particle generation model is conditioned on the overall jet kinematics described by $(p_{Tjet}, \eta_{jet} mass, N_{part})$ The overall jet kinematic information is learned (simultaneously) by a second diffusion model as done in Ref. [42] using a model based on the RESNET [66] architecture.

All features are first normalized to have mean zero and unit standard deviation before training. The probability density is calculated with Eq. 3. The integral is solved using SCIPY [67] with explicit Runge-Kutta method of order 5(4) [68,69] with absolute and relative tolerances of $5 \times 10^{-5}$ and $10^{-4}$, respectively. Lower and higher values of the absolute and relative tolerances were tested with overall results remaining unchanged.

First, we demonstrate the permutation invariance of the probability density by evaluating the estimated negative log-likelihood (nll) of the data, trained using exclusively QCD jets. We show a single jet using different permutations of the input particles. These results are presented in Figure. 1. Uncertainties are derived from the standard deviation of 10 independent estimations of the nll.

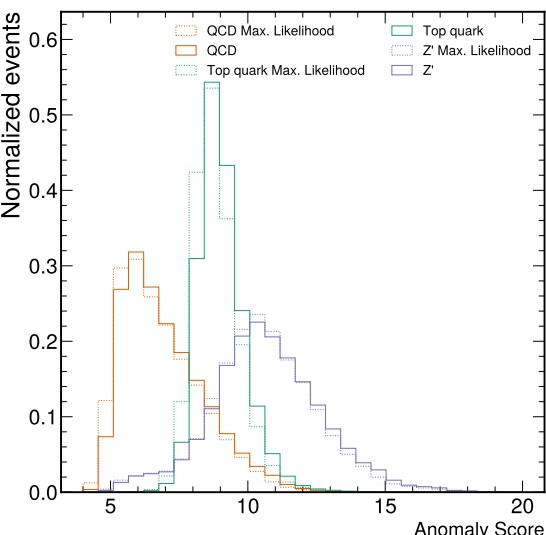

Figure 2: Anomaly score for QCD, top quark, and $Z'$ jets evaluated on the model trained exclusively on QCD jet events.

Since the model was trained only on QCD jet events, the estimated nll tends to be lower for QCD jets compared to the other classes. This observation motivates the use of the nll as an anomaly score to identify jets with low likelihood. On the other hand, the varying particle multiplicity makes the comparison between jets with different number of constituents misleading. Since the densities are expected to be correctly normalized for each fixed value of the particle multiplicity, jets with higher number of particles will yield low probability densities regardless of the sample used during training. To account for this issue, we define the anomaly score as

$$\text{anomaly score} = -\log(p(\text{jet})\, p(\text{part}|\text{jet})^{1/N}), \tag{7}$$

with the model learning the likelihood in the particle space conditioned on the jet kinematic information ($p(\text{part}|\text{jet})$) normalized by the particle multiplicity.

We show the distribution of the anomaly score in Fig. 2 for diffusion models trained exclusively on QCD jets and provide the distributions of the nll without the normalization factor in App. A.

The diffusion model training using maximum likelihood ($\lambda(t) = g(t)^2$) also presents, on average, lower anomaly score compared to the standard diffusion approach ($\lambda(t) = \sigma(t)^2$). With this choice of anomaly score, we investigate the the significance improvement characteristic curve (SIC), shown in Fig. 3.

For both classes of anomalies we observe maximum values for the SIC curve above 1, supporting the choice of metric for anomaly detection. Conversely, the maximum-likelihood training results in slightly lower SIC curve for anomalous jets containing the decay products of top quarks. Similarly, we can train the diffusion model on a dataset containing only top quark initiated jets and evaluate the estimated anomaly score using different jet categories. The result is shown in Fig. 4. In this case, the anomaly score values for top quark initiated jets are lower on average compared to the other categories.

A key challenge with unsupervised AD is how to compare different methods. Weakly supervised methods based on likelihood ratios can be compared with an optimal classifier using the same noisy label inputs [70, 71] and they converge to the fully supervised classifier in the limit of large signal, independent of the signal properties. Unfortunately, there is no analog

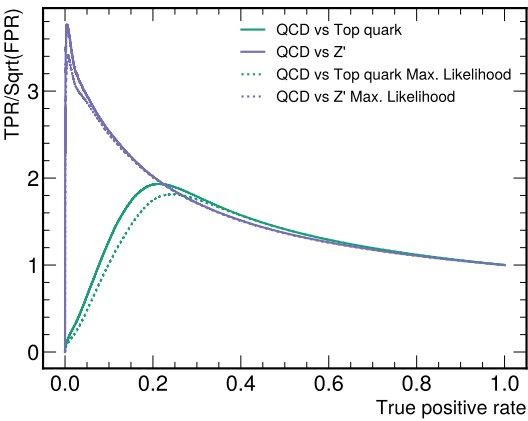

Figure 3: Significance improvement characteristic curve for different classes of anomalies investigated in this work.

for this in the unsupervised case. The existing papers on unsupervised AD compare methods by demonstrating the performance on a (usually small) set of benchmark signal models, as we have also done in Fig. 3. However, this is a model-dependent comparison whose conclusions can easily be altered by simply changing the physics model(s) considered [20]. As the unsupervised AD hypothesis is that the new physics, if it exists, is rare given some set of coordinates, then one could instead directly compare the fidelity of the density estimation in the background-only case. Since the true probability density is unknown, this can be achieved using likelihood-ratio methods.

Recent studies have used classifier-based likelihood ratio estimation to assess and/or improve deep generative models [31–34, 36, 40, 72–75]. These classifiers are trained using samples drawn from the generative model and from the target dataset. As with the training of a Generative Adversarial Network (GAN) [76], when the classifier is not able to distinguish the generative model and the target, then the generative model is accurate. Density estimators are a subclass of generative models and could be evaluated in this way. However, being able to effectively produce samples and being able to estimate the probability density are often at odds with each other and so it is desirable to have a comparison method that uses the probability density without relying on sampling.

Following Ref. [30], we use another approach that directly assesses the quality of explicit density-based AD methods. Given two samples (e.g. top quark and QCD jets), we take the ratio of learned densities (see also Ref. [70]) and compare the resulting score to a fully supervised classifier trained to distinguish the same two samples. The likelihood ratio is the optimal classifier [77] and if the density estimation is exactly correct and the classifier is optimal, then these two approaches should have the same performance. Training a supervised classifier is an easier problem (Ref. [70] versus Ref. [71]), so a reasonable approximation is that the classifier can be made nearly optimal. For the top-tagging dataset, this problem has already been studied extensively (see e.g. Ref. [50] and papers that cite it). This approach does depend on the samples used to estimate the likelihood ratio, but it is still a sensitive test to the density across the phase space. In Fig. 5, we calculate the receiver operating characteristic (ROC) curves obtained in the anomaly detection task using the anomaly score metric (Eq. 7) and evaluated on samples where either QCD or top quarks are considered the main background. We also provide the ROC curves obtained using the log-likelihood ratio between two dedicated diffusion models, trained exclusively on QCD or top quark jets, and the one obtained from the outputs of a classifier. The classification network is trained using the same network architecture

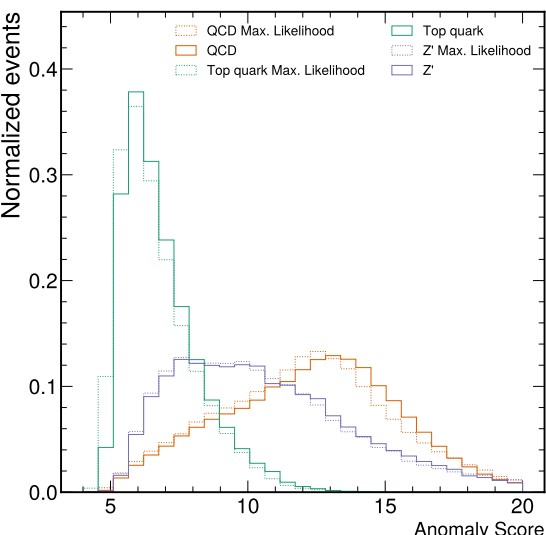

Figure 4: Anomaly score for QCD, top quark, and $Z'$ jets evaluated on the model trained exclusively on top quark jet events.

as the diffusion model for particle generation, with additional pooling operation after the last transformer layer, followed by a fully connected layer with LEAKYRELU activation function and 128 hidden nodes. The output of the classifier is a single number with a SIGMOID activation function.

The ROC curve obtained using the log-likelihood ratio has similar area under the curve (AUC) as the dedicated classifier, even though the performance still differs significantly in the whole true positive range. Similar results are found in Ref. [30]. This suggests that even though we are using a state-of-the-art density estimation strategy, there is still plenty of room to innovate in order to close the performing gap. Additionally, this illustrates the danger of relying only on AUC, since it may not be sensitive to tails of phase space relevant for AD. Similarly to the previous study, we only observe marginal differences between the results obtained from the different strategies used to train the diffusion model.

In Table 1, we present a summary of the results consisting of the maximum SIC value, AUC for the anomaly detection task and supervised density estimation.

## 5 Conclusions and outlook

In this work we presented an unsupervised anomaly detection methodology based on diffusion models to perform density estimation. Our method approximates the score function to estimate the probability density of the data. The diffusion model is trained directly on low-level objects, represented by particles clustered inside jets. The model for the score function is equivariant with respect to permutations between particles, leading to a permutation invariant density estimation. We test different strategies to train the diffusion model, including a standard implementation and a maximum-likelihood training of the score model. The maximum-likelihood training presents on average a lower negative-log-likelihood, indicating improved probability density estimation. However, when applied for anomaly detection, we do not observe notable improvements.

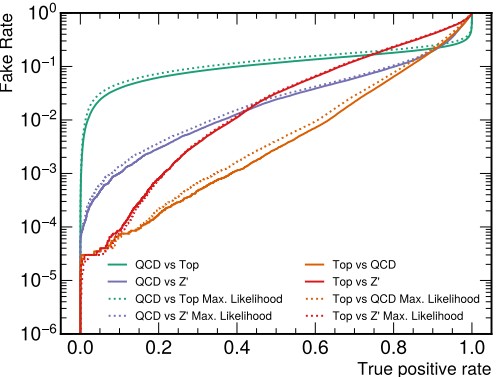 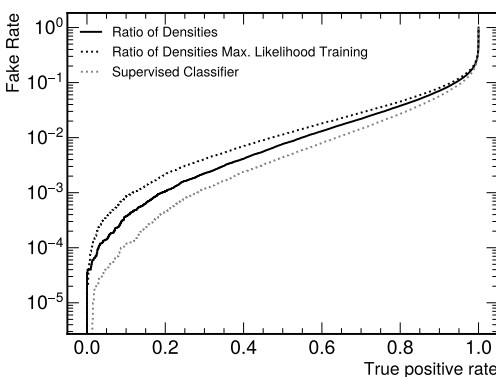

Figure 5: Receiver operating characteristic curve obtained from the unsupervised anomaly detection strategy (left), direct density estimation, and a supervised classifier (right). The density ratio uses the estimated densities from individual diffusion models trained either with QCD or top quark jets. The classifier is trained to separate QCD from top quark jets.

Table 1: Comparison of different quality metrics for the anomaly detection task using different datasets. We consider both the scenarios where the model is trained on QCD (QCD vs.) or top quarks (Top vs.) and is evaluated against other physics processes not used for training. Results are reported using the standard diffusion training with maximum-likelihood training results in parenthesis. For comparison, we present the AUC obtained from the classification of top quarks from QCD jets using the ratio of the estimated densities, or directly training a classifier on the same dataset. Bold quantities represent the best model for a given metric.

| Dataset | Max. SIC | Unsupervised AUC | Density Ratio AUC | Sup. AUC |
|---|---|---|---|---|
| QCD vs. Top | **1.93** (1.81) | **0.875** (0.855) | 0.975 (0.971) | 0.980 |
| QCD vs $Z'$ | **3.76** (3.42) | **0.924** (0.919) | - | - |
| Top vs. QCD | **16.0** (15.4) | **0.937** (0.930) | - | - |
| Top vs $Z'$ | 11.0 (**12.5**) | **0.875** (0.872) | - | - |

Additionally, we evaluate the density estimation performance by studying the log-likelihood ratio for two density estimators; one trained on QCD jet events and the other exclusively on top quark jet events. The dedicated classifier shows a better performance compared to the individual estimation of the log-likelihood ratio, indicating room for improvement.

For future studies, we plan to investigate alternative diffusion strategies beyond our implementation to improve the density estimation. Those include high-order denoising score-matching [78] or using a learnable reweighing scheme presented in Ref. [49], both showing promising density estimation performance. There may also be additional applications of high-dimensional, permutation-invariant density estimation beyond anomaly detection.

## Acknowledgments

We thank Julia Gonski, Manuel Sommerhalder, Michael Krämer, and Thorben Finke for feedback on the manuscript.

**Funding information** VM and BN are supported by the U.S. Department of Energy (DOE), Office of Science under contract DE-AC02-05CH11231. This research used resources of the National Energy Research Scientific Computing Center, a DOE Office of Science User Facility supported by the Office of Science of the U.S. Department of Energy under Contract No. DE-AC02-05CH11231 using NERSC award HEP-ERCAP0021099.

**Code availability** The code for this paper can be found at https://github.com/ViniciusMikuni/PermutationInvariantAD.git.

# A  Negative log-likelihood distributions

We introduced the anomaly score used in this work as a function of the likelihood obtained from the jets. In this section we show the distributions of the negative log-likelihood (nll) without modifications as we estimate from the trained models. In Fig. 6, we show the nll for a model trained exclusively on QCD jets (left) or exclusively on top quark initiated jets (right). We observe that, without the normalization factor, QCD jets are always identified with lower nll while top-quark initiated jets always present higher nll and would be considered anomalous in both training scenario.

# B  Comparison with previous publications

In our studies, we used a dataset consisting of semi-visible jets. Similarly, in [63], semi-visible jets using different PYTHIA settings were used to evaluate the performance of the anomaly detection task. To enable comparisons between our proposed model and previous results we show in Tab. 2 using a dataset that, to the best of our ability, matches the one used in [63]

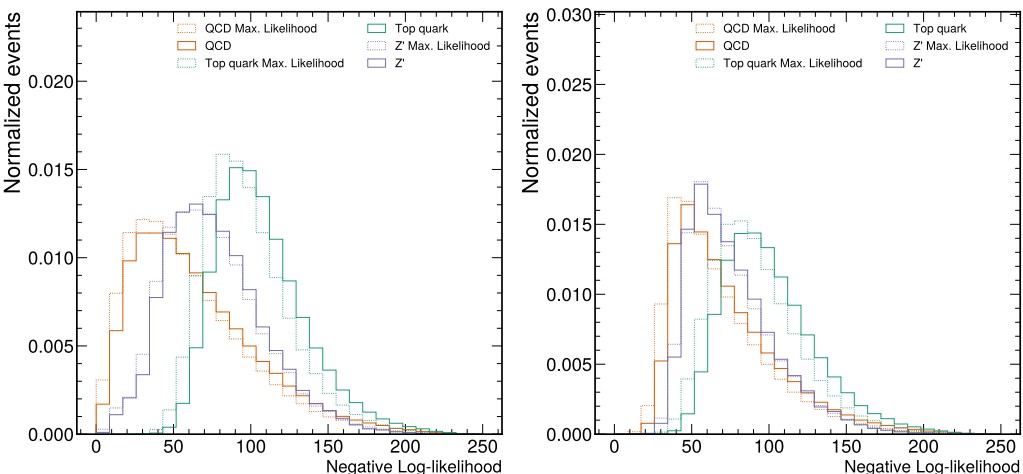

Figure 6: Negative log-likelihood for QCD, top quark, and $Z'$ jets evaluated on the model trained exclusively on QCD jets (left) or top quark jets (right).

Table 2: Comparison of different quality metrics for the anomaly detection task. We consider both the scenarios where the model is trained on QCD (QCD vs.) or top quarks (Top vs.) and is evaluated against the dark shower sample not used during training.

| Dataset | Max. SIC | Unsupervised AUC |
|---------|----------|------------------|
| QCD vs $Z'$ | 3.4 | 0.71 |
| Top vs $Z'$ | 7.0 | 0.93 |

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
