# Peer review of "High-dimensional and Permutation Invariant Anomaly Detection"

_SciPost Physics, doi:SciPost Phys. 16, 062 (2024)_

## Round 3 · Referee Report · Anonymous (Referee 1) · 2023-11-6

Strengths

1 -The paper details a novel computational method for using the learned density as a permutation-invariant anomaly detection score to improve anomaly detection. The authors show that the method is effective at dealing with higher dimensional parameter spaces.

2-It is written in a clear and intelligible way, it contains a good description of the problem and objectively evaluates the method against other methods

3-Code is provided to enable reproducibility. It contains a clear conclusion summarizing the results perspectives for future work.

Weaknesses

None

Report

The paper is a well written comprehensive study and I find it to be acceptable in its present form.

Requested changes

None

---

## Round 3 · Referee Report · Tilman Plehn (Referee 2) · 2023-11-22

Report

The paper presents a seriously interesting application of modern neural networks to a key task at the LHC. It is novel, relevant, and well written. Sorry for being late with my report. I only have a few questions/remarks for the authors to consider, largely voluntarily:

  • At the top of p.2 you say that there are no applications of diffusion models for density estimation. I view generative diffusion models over phase space as density estimation, but I could be easily convinced to agree with a more specific statement...
  • As I have said for other papers, any chance you could include some kind of graphic representation of your network setup? So people can use it during talks?
  • Why do you modify our two dark-jet datasets? I know it is extra work, but I think it would be very useful to show results for our Aachen' andHeidelberg' datasets. Any chance you could add that, to see what your network does when challenged more seriously?
  • I am sorry, but I do not understand the argument before Eq.(7). The number of constituents is also one of the most useful observables to find semi-visible jets. What is it exactly that you want to achieve with this anomaly score in relation to N? And N = N_part? Sorry for not getting your point.
  • Could you maybe add some more information to Tab.1, for instance on the inverse, Top background vs QCD signal performance? As far as I can see this table is the only way to compare your results to, for instance, Fig.4 in `QCD or What' or Fig.4 in the NAE paper.
  • Similarly, any change you could show the inverse QCD signal performance in Fig.5? Do the ROC curves look the same?
  • I am a little lost in your conclusions. In the beginning, your paper seems to be about the unsupervised density estimation, but the conclusion then ends with the positive note on the density ratio?

---

## Round 5 · Author Response

The authors would like to thank the referees for the insightful comments made during the review process. We incorporated the feedback in the text and give detailed answers below to comments received.

The paper presents a seriously interesting application of modern neural networks to a key task at the LHC. It is novel, relevant, and well written. Sorry for being late with my report. I only have a few questions/remarks for the authors to consider, largely voluntarily:

  • At the top of p.2 you say that there are no applications of diffusion models for density estimation. I view generative diffusion models over phase space as density estimation, but I could be easily convinced to agree with a more specific statement…

Even though the generation step for diffusion models implicitly sample from the density, you normally don’t have direct access to the value of the density from the trained model unless you solve the associated ODE. We mention that our application is the first explicit method for density estimation with low level inputs to make this distinction more clear. We also added to the text to make the point more clear: Similarly, generative models are capable to implicitly learn the the density through data generation. In this case, high quality samples can be created but the exact value of the density is not explicitly known.

  • As I have said for other papers, any chance you could include some kind of graphic representation of your network setup? So people can use it during talks?

Our model uses the same architecture as the one we introduced in the fast point cloud generative model published in Phys. Rev. D 108, 036025. There we provide the network setup in Fig. 2

  • Why do you modify our two dark-jet datasets? I know it is extra work, but I think it would be very useful to show results for our Aachen and Heidelberg' datasets. Any chance you could add that, to see what your network does when challenged more seriously? In our initial studies we have used similar settings to the samples used in the dark-jets paper. However, at the time, we observed some issues in the performance evaluated over these samples, which were fixed when we improved the likelihood estimation of our model. Nevertheless, at the time we had already changed the settings of the dataset we were using to evaluate the model performance. Since there are no public repositories containing the dark jets dataset, we have, to the best of our ability, tried to reproduce the dataset from the dark-jets publication and provide an appendix with the results of the ROC and maximum SIC curves for this dataset.

  • I am sorry, but I do not understand the argument before Eq.(7). The number of constituents is also one of the most useful observables to find semi-visible jets. What is it exactly that you want to achieve with this anomaly score in relation to N? And N = N_part? Sorry for not getting your point.

Indeed, the number of constituents is an important feature. The problem we faced is that comparing jets with different multiplicities in terms of the density is tricky since the dimensional space where the densities are calculated is not the same (note that density has dimensions!). Our solution was to use the particle multiplicity to normalize the particle density model (p(part|jet)), such that the difference in dimensionality, irrespective to the other properties of the particles, is not the deciding factor when comparing the densities. We still include the particle multiplicity in the jet density model, p(jet), as one of the features such that we are still sensitive to this feature.

  • Could you maybe add some more information to Tab.1, for instance on the inverse, Top background vs QCD signal performance? As far as I can see this table is the only way to compare your results to, for instance, Fig.4 in `QCD or What' or Fig.4 in the NAE paper. We have also included the values of the maximum SIC curve and AUC for Top quarks vs. QCD and the Z’ sample.

  • Similarly, any change you could show the inverse QCD signal performance in Fig.5? Do the ROC curves look the same? We have now added the additional ROC curves for top vs QCD and top vs Z’ to figure 5 left.

  • I am a little lost in your conclusions. In the beginning, your paper seems to be about the unsupervised density estimation, but the conclusion then ends with the positive note on the density ratio? In this paper we also introduced the comparison of the likelihood ratio, estimated from the ratio of learned densities, with the estimation derived from the results of a trained classifier. Based on Figure 5 right we see that currently, the density estimator produces a worse ROC curve compared to the classifier. This comparison is important since without knowledge of the true likelihood, determining how good the density estimation is becomes non-trivial.

---

## Round 5 · List of Changes

• Textual changes to address the referee's comments: Added: Similarly, generative models are capable to implicitly learn the the density through data generation. In this case, high quality samples can be created but the exact value of the density is not explicitly known.

  • Added a new appendix comparing the result of our model with previously used dark jets datasets -Added the maximum SIC curve and AUC for Top quarks vs. QCD and the Z’ sample as a table.

  • Added the additional ROC curves for top vs QCD and top vs Z’ to figure 5 left.

---

## Editorial Decision

published